# Pomegranate Peel and Olive Leaf Extracts to Optimize the Preservation of Fresh Meat: Natural Food Additives to Extend Shelf-Life

**DOI:** 10.3390/microorganisms12071303

**Published:** 2024-06-27

**Authors:** Giuseppina Forgione, Giuseppa Anna De Cristofaro, Daniela Sateriale, Chiara Pagliuca, Roberta Colicchio, Paola Salvatore, Marina Paolucci, Caterina Pagliarulo

**Affiliations:** 1Department of Science and Technology, University of Sannio, via F. De Sanctis Snc, 82100 Benevento, Italy; gforgione@unisannio.it (G.F.); gadecristofaro@unisannio.it (G.A.D.C.); sateriale@unisannio.it (D.S.); paolucci@unisannio.it (M.P.); 2Department of Molecular Medicine and Medical Biotechnologies, University of Naples Federico II, via S. Pansini 5, 80131 Naples, Italy; chiara.pagliuca@unina.it (C.P.); roberta.colicchio@unina.it (R.C.); psalvato@unina.it (P.S.); 3CEINGE-Biotecnologie Avanzate s.c.ar.l., via G. Salvatore 486, 80145 Naples, Italy

**Keywords:** pomegranate peel, olive leaves, antimicrobial activity, shelf-life extension, biochemical preservation, meat quality

## Abstract

Quality and safety are one of the main concerns of the European Union in food preservation. Using chemical additives extends the shelf-life of fresh foods but raises consumer’s concerns about the potential long-term carcinogenic effects. Using natural substances derived from agro-industrial by-products, which have significant antimicrobial and antioxidant activities, could extend the shelf-life of fresh foods such as meat. Furthermore, they can provide nutritional improvements without modifying organoleptic properties. This study analyzes the antimicrobial activity of pomegranate peel extract (PPE) and the antioxidant activity of olive leaf extract (OLE), added at concentrations of 10 mg g^−1^ and 0.25 mg g^−1^, respectively, to minced poultry and rabbit meat. PPE exhibited in vitro antimicrobial activity against foodborne pathogens starting at 10 mg/well. PPE and OLE determined a reduction in colony count over a storage period of 6 days at 4 °C. Additionally, the combination of PPE and OLE showed antioxidant effects, preserving lipid oxidation and maintaining pH levels. The obtained results demonstrate that PPE and OLE can be recommended as food additives to preserve the quality and extend the shelf-life of meat products.

## 1. Introduction

Ensuring the safety and quality of fresh foods, particularly perishable items like meat, is a critical concern in the modern food industry. While fresh foods are rich in nutrients and contribute to a balanced diet, they also pose inherent risks of foodborne infections and intoxications [1]. Fresh foods, particularly meat, are highly susceptible to microbial contamination and subsequent quality deterioration. The natural composition of meat provides an ideal environment for the proliferation of microorganisms, leading to changes in color, texture, and flavor [2,3,4].

The European Union regulates self-control throughout all stages of the food chain to ensure the safety and quality of food and consumer’s well-being. In particular, Regulation (CE) n° 2073/2005 [5] establishes microbiological criteria to determine whether food is satisfactory, acceptable, or unacceptable. In this case, the concept of shelf-life is crucial, especially for meat products whose quality and safety can be significantly influenced by microbial growth and activity. Managing and extending the shelf-life of perishable foods, therefore, involves strategies to control and reduce microbial growth. In this context, understanding the dynamics of microbial growth and the behavior of microorganisms in the presence of natural substances in meat products is essential for developing effective preservation methods. The use of chemical additives may help to prolong the shelf-life of these foods but the long-term addition of such substances could have negative effects on consumer health [6,7]. In this context, the use of natural substances with antimicrobial and antioxidant properties as food additives that can also enhance the nutritional properties of foods is of relevant interest. Many natural substances are derived from agri-food waste chains such as olive leaves, grape vines, or pomegranate peel [8,9,10].

Pomegranate peel contains many bioactive compounds, especially polyphenols like tannins and flavonoids [10]. Pomegranate, recognized as a safe food by the European Food Safety Authority [11], is a nutraceutical with diverse health benefits. It exhibits antidiabetic properties, regulates blood glucose levels, prevents cardiovascular diseases, reduces blood cholesterol levels, demonstrates anticarcinogenic activity, protects lipids from oxidative damage, and possesses antioxidant and anti-aging properties, particularly beneficial for brain health [11,12,13]. Also interesting is the recent demonstration of the antinociceptive effect of bioactive compounds from pomegranate peel and juice in an animal model of pain, highlighting the potential of pomegranate as a natural alternative for the treatment of nociceptive and inflammatory pain [14]. Scientific studies have also highlighted the significant antimicrobial properties found in these compounds, leading to a reassessment of the peel’s potential applications [10,15,16]. The antimicrobial activity, demonstrated by the ability of pomegranate peel extracts to hinder the growth of harmful microorganisms, is crucial for food preservation. The integration of pomegranate-derived compounds in meat preservation has gained significant attention due to their potential to reduce microbial activity and enhance the shelf-life of meat products. The natural antimicrobial properties of the bioactive components of pomegranate offer a promising solution to diminish microbial proliferation in the meat [17].

Foods like meat—rich in proteins and lipids and nutrient-dense—are also prone to lipid oxidation and protein degradation due to microbial contamination. The formation of free radicals is linked to typical aerobic metabolism. Oxygen consumption during cell growth generates a range of oxygen free radicals. When these species interact with lipid molecules, they produce new radicals, hydroperoxides, and various peroxides, which lead to rapid biological deterioration during meat processing and storage. Important antioxidant properties are attributed to natural extracts, in particular, olive leaf is rich in secoiridoids (such as oleuropein and hydroxyoleuropein), alcohol phenols (including hydroxytyrosol and tyrosol), hydroxycinnamic acid derivatives (like verbascoside), and flavonoids (such as apigenin-7-O-glucoside and diosmetin-7-O-glucoside), which exhibit antioxidant properties against superoxide, hydroxyl, and peroxide radicals—primarily attributed to the redox properties of their phenolic hydroxyl groups [18,19,20]. Several studies demonstrated the beneficial effect of olive leaf extract on human health and highlighted its antioxidant effect in different food matrices, such as oils, meat, baked goods, vegetables, and dairy products [21]. Rubel et al. incorporated olive leaf extract into ground meatballs and observed that this enhances the organoleptic and nutraceutical properties of the product, significantly reducing oxidative damage to fats and proteins [22].

Using natural extracts as food preservatives with antimicrobial and antioxidant properties can extend the shelf-life of perishable products such as white meat and even enhance their organoleptic and nutraceutical properties. In particular, the incorporation of pomegranate peel and olive leaf extracts with high bioactive properties not only enhances the quality and shelf-life by inhibiting microorganisms’ growth and oxidative deterioration of proteins and lipids but also augments the functional and health-promoting aspects of foods [23]. Recent studies confirm the use of olive leaf extract as an antioxidant in poultry meat with satisfactory effects without altering the organoleptic properties of the product [24,25], while the use of pomegranate peel extract on white meats, such as poultry and rabbit, was not thoroughly evaluated [25,26]. This study aims to analyze the use of pomegranate peel and olive leaf extracts as natural additives to extend the shelf-life of perishable products such as poultry and rabbit minced meat without altering the organoleptic and nutritional properties of the products.

## 2. Materials and Methods

### 2.1. Plant Materials

Pomegranate fruit (*Punica granatum* L.) was harvested from small farms in the Campania region (Italy). The fruit underwent meticulous manual cleaning, and the peel was delicately removed. Subsequently, the pomegranate peel was air-dried for several days and finely pulverized.

Olive leaves (*Olea europaea* L.) were kindly provided by Oleificio Vascello (Morcone, BN, Italy) during the olive harvesting period (November). The leaves were first air-dried for several days and then finely powdered.

### 2.2. Preparation of Polyphenolic Extracts

The hydro-alcoholic polyphenolic extract from pomegranate peel (PPE) was prepared following the method described by Pagliarulo et al. [10]. In brief, pomegranate peel was mixed with ethanol and water in a 1:1 ratio (*v*/*v*) for 30 min in darkness, resulting in a final concentration of 0.2 g mL^−1^. The extract was first filtrated using Whatman CHR 1 (Life Sciences, Milano, Italy) and the volume was reduced using a rotavapor (Heidolph 36001270 Hei-VAP Precision Rotary Evaporator). The remaining portion was subsequently freeze-dried.

The olive leaf extract (OLE) was obtained through microwave-assisted extraction as reported in De Cristofaro et al. [26]. A total of 1 g of powdered olive leaf was first mixed by agitation at 150 rpm in 25 mL of distilled water (1:25 ratio), and then subjected to microwave extraction for 2 min at 300 W. Thereafter, the extract was filtered with Whatman CHR 1 (Life Sciences) and then frozen at −80 °C and freeze-dried to obtain a fine powder.

Before use, the freeze-dried powders were reconstituted in distilled water and filtered using Millex-GS filters (Merk-Millipore, Darmstadt, Germany) with a porosity of 0.45 μm.

### 2.3. In Vitro Antimicrobial Activity of Pomegranate Extract

#### 2.3.1. Selected Microorganisms for In Vitro Tests

Preliminary antimicrobial tests were conducted using American Type Culture Collection (ATCC) strains, kindly provided by the Istituto Zooprofilattico Sperimentale del Mezzogiorno (Portici, NA, Italy). Specifically, in vitro assays were performed against *E. coli* ATCC 25922 clinical isolate used for food testing; *S. aureus* ATCC 25923 clinical isolate usable for research on infectious and enteric diseases; *B. cereus* ATCC 14579 for testing and food production research; and *S. enterica* ATCC 14028, isolated from chicken heart and liver tissue used for food tests. All strains were cultured on Luria Bertani (LB) agar medium (12780-029, Invitrogen, Waltham, MA, USA) and incubated at 37 °C for 24 h to ensure their viability for performing the assays.

#### 2.3.2. Agar-Well Diffusion Method

To evaluate the in vitro antibacterial properties of PPE against specific microorganisms, an antimicrobial activity test was conducted using the agar-well diffusion method, as reported by Perez et al. [27], with some modifications. In brief, bacterial strains were cultured in LB broth, and aliquots of each microbial suspension with a 600 nm optical density of 0.5 O.D. were evenly spread on Petri dishes containing LB agar medium. Subsequently, wells with a diameter of 6 mm were created using sterile glass Pasteur pipettes, and aliquots of PPE were introduced into the wells. The plates were then incubated under aerobic conditions at 37 °C for 24 h. Then, the diameter (expressed in mm) of the inhibition zones was measured, and the antibacterial activities were presented as the mean diameter of the inhibition zones (MDIZ) produced by PPE against the tested microorganisms. The antibiotics gentamicin (Sigma-Aldrich S.r.l., Milano, Italy), vancomycin (Gold-biotechnology, Saint Louis, MI, USA), and amoxicillin (Aesculapius Farmaceutici S.r.l., Brescia, Italy) were used as positive controls, while the extraction buffer was used as a negative control. The experiments were conducted in triplicate with independent cultures.

#### 2.3.3. Tube Dilution Method

The susceptibility of microorganisms to increasing concentrations of PPE was determined through a broth dilution method, starting from a standard inoculum of 1 × 10^5^ CFU mL^−1^ (Colony-Forming Units mL^−1^), following the Clinical and Laboratory Standards Institute (CLSI) 2022 guidelines [28]. In brief, various concentrations of PPE were directly added to the LB broth medium. Adequate suspension of the extract within the broth was maintained by vigorous agitation using a vortex mixer and continuous shaking during incubation. This method allowed the quantitative evaluation of the antibacterial effects of PPE by determining the minimum inhibitory concentration (MIC) and the minimum bactericidal concentration (MBC). MIC and MBC values were determined by incubating bacterial cultures with PPE at increasing concentrations (0, 2.5, 5, 10, 20, 40, 60, 80, 100 mg mL^−1^) under appropriate growth conditions with continuous agitation. Subsequently, the observation of tube turbidity was conducted, and aliquots were plated on LB agar medium and then incubated at 37 °C for 24 h. MIC was defined as the lowest concentration of an antibacterial agent that prevents bacterial growth in vitro, while MBC was designated as the minimum concentration of an antibacterial agent that kills 99% of the bacteria from the initial inoculum in vitro. Gentamicin, vancomycin, and amoxicillin were used as positive controls, while the extraction buffer was used as a negative control. The experiments were conducted in triplicate with independent cultures.

### 2.4. Shelf-Life Analysis of Minced Poultry Meat and Minced Rabbit Meat

#### 2.4.1. Experimental Design and Samples Preparation

The shelf-life assessment analyses were conducted on poultry (P), yellow poultry (YP), and rabbit (R) minced meat produced and marketed by the “Avicola Mauro S.r.l.” company (Paolisi, BN, Italy). Samples were treated with a polyphenol mixture composed of 10 mg of PPE and 0.25 mg of OLE for a gram of meat. PPE and OLE were blended with the meat using sterile L-shaped loops. Control samples (CT) were P, YP, and R without PPE and OLE. In summary, the groups were as follows: P-CT (poultry control or untreated); YP-CT (yellow poultry control or untreated); R-CT (rabbit control or untreated); P-TR (poultry treated with polyphenol mixture); YP-TR (yellow poultry treated with polyphenol mixture); R-TR (rabbit treated with polyphenol mixture). The microbiological quality of the minced meat was determined after 0, 3, and 6 days of storage. On the delivery day, the meat samples were treated with polyphenolic extract and stored at 4 °C, except for those from day 0, which were immediately analyzed.

#### 2.4.2. Food Microbiological Testing

The qualitative and quantitative assessment of the microbial load in P, YP, and R was performed using culture-dependent techniques. Twenty-five-gram portions of meat were collected according to standard aseptic procedures. The collected samples were transferred into sterile polyethylene bags (Whirl-Pak™, Wisconsin, USA), then homogenized in Peptone Water (Thermo Scientific Oxoid, Hampshire, UK) at a 1:10 ratio using a peristaltic homogenizer Stomacher 400 Circulator (Seward Ltd., Worthing, UK). At each analysis time, and for each sample, four homogenization cycles were conducted, each lasting 30 s at 230 revolutions per minute. Then, appropriate dilutions of the homogenate were prepared using the method of serial dilutions in Peptone Water. Once decimal dilutions of the samples were prepared, aliquots were inoculated onto selected nutrient media. Specifically, the targeted microorganisms and incubation conditions were as follows: total aerobic mesophilic bacterial count on LB Agar (Invitrogen, Thermo Fisher Scientific Inc.) at 37 °C under aerobic conditions for 48 h; Enterobacteriaceae on MacConkey Agar (Merck, Darmstadt, Germany); *E. coli* β-glucuronidase positive on Tryptone Bile X-Glucuronide Chromogenic Agar (Merck, Darmstadt, Germany) at 37 °C under aerobic conditions for 48 h; *Pseudomonas* spp. on Cetrimide Agar Base (Conda, Madrid, Spain) at 37 °C under aerobic conditions for 48 h; coagulase-positive Staphylococci on Baird Parker Agar (Merck, Darmstadt, Germany) at 37 °C under aerobic conditions for 48 h; anaerobic sulfite-reducing bacteria (*Clostridium* spp.) on Tryptose Sulfite Cycloserine Agar (Merck, Darmstadt, Germany) for 48 h at 37 °C under anaerobic conditions, created using an AnaeroJar anaerobic jar (Oxoid Ltd., Hampshire, England) and bags for the Atmosphere Generation System CampyGenTM (Thermo Fisher Scientific); yeasts on Potato Dextrose Agar (Merck, Darmstadt, Germany) at room temperature (25 °C) under aerobic conditions for 96 h. Upon completion of growth, colony counts were determined, and the number N of CFU g^−1^ of meat was calculated using the formula:(1)$$N=\frac{\sum C}{V}\cdot d$$
where ΣC is the sum of colonies counted on multiple plates obtained from successive dilutions, with at least one containing a minimum of 10 colonies; V is the volume of the inoculum seeded on plates; d is the dilution factor corresponding to the considered dilution. All experiments were conducted in triplicate for each established storage period. The results obtained were expressed in Colony-Forming Units (CFU) per gram of product and compared with the acceptability levels set by Reg. CE 2073/2005 for refrigerated minced meat [5].

#### 2.4.3. Antioxidant Activity

The free radical scavenging capacity was assessed through the 2,2-diphenyl-1-picrylhydrazyl (DPPH) assay, as reported in Ramli et al. [29]. In brief, 0.03 g of meat was continuously mixed with 3 mL of 0.004% DPPH (Sigma-Aldrich S.r.l., Milano, Italy) solution in methanol (Sigma-Aldrich S.r.l., Milano, Italy) for 30 min in the dark. The samples were then centrifuged at 1430× *g* for 10 min. Finally, using methanol as a blank, the absorbance of the supernatant was measured at 517 nm. The percentage of oxidative inhibition was calculated according to the following equation:(2)$$Oxidative\;inhibition\;\%=\frac{\left(A_{DPPH}-A_{sample}\right)}{A_{DPPH}}\cdot 100$$
where A_DPPH_ and A_sample_ are the absorbance at 517 nm of the pure DPPH solution and the sample after the test, respectively. The measurement was performed on both raw and microwave-cooked meat samples (300 W for 50 s). The experiment was performed in triplicate on three independent samples.

#### 2.4.4. Lipid Peroxidation

The thiobarbituric acid assay (TBARS) is one of the most used methods for assessing lipid oxidation. The test was performed as reported in Fasseas et al. [30] with some modifications. Briefly, 0.03 g of meat was mixed with a solution containing 0.6 mL of dH_2_O, 0.9 mL of phosphoric acid (pH 2) (Supelco, Darmstadt, Germany), and 0.9 mL of 0.8% (*w*/*w*) 2-Thiobarbituric acid (TBA) (Sigma-Aldrich, Darmstadt, Germany) diluted in 1.1% sodium dodecyl sulfate (SDS) (Sigma-Aldrich, Darmstadt, Germany). The sample was vortexed and heated at 100 °C for 60 min in a thermoblock (Block Heater Stuart SBH130D/3). Subsequently, the sample was cooled in ice for 5 min, and 3 mL of 1-butanol (Sigma-Aldrich, Darmstadt, Germany) was added. The sample was then mixed and centrifuged at 8900× *g* for 10 min. The absorbance was read at 532 nm using butanol as blank. The sample’s absorbance was interpolated with the standard calibration curve of malondialdehyde (MDA), and the results were expressed as nmol of MDA. The experiment was performed in triplicate on three independent samples.

#### 2.4.5. pH Measurement

The pH variations were measured before and after microwave cooking, following the procedure outlined in Ramli et al. [29]. In brief, 0.5 g of meat was homogenized with 3 mL of distilled water using Ultraturrax, and the homogenate was centrifuged at 2240× *g* for 5 min. The supernatant was filtered, and the pH of the solution was measured using a pH meter (pH 50 VioLab). The measurement was performed on both raw and microwave-cooked meat samples (300 W for 50 s).

#### 2.4.6. Colorimetric Assay

The study of color variation was conducted using a Computer Vision System, as reported by Tomasevic et al. [31], with some modifications. In short, a 12 MPX digital camera (Sony DSLR-A500L) was used, positioned vertically at a distance of 30 cm from the sample, with the following settings: shutter speed 1/6 s, manual mode, focal aperture 11.0, ISO speed 100, flash deactivated, focal length 30 mm. Illumination was provided by a strip of LED lights with a color temperature of 4500 K (neutral white). To achieve uniform light intensity without shadows on the sample, the LEDs were positioned above the samples. Both the LED light and the camera were fixed inside a box (side length = 50 cm) with internal walls coated with opaque black fabric to attenuate background light. Before capturing the images, the camera was calibrated using a 24-color Colorchecker. Adobe Photoshop CC 2021 software was used for image analysis. The colorimetric features of RGB images were acquired using RAW photographs. Three photos of each sample were performed, and as it was not possible to assess the color over the entire surface of the sample, seven readings for photos (technical replicates) were randomly chosen for measurements, and colorimetric parameters of brightness (L^●^), redness (a^●^), and yellowness (b^●^) were measured.

#### 2.4.7. Statistical Analysis

The statistical significance of the data was analyzed with GraphPad Prism version 8.0.2 (GraphPad, Inc., San Diego, CA, USA) and results are expressed as mean ± SD. Data obtained by agar-well diffusion method (Section 2.3.2) were analyzed by one-way ANOVA test with Dunnett’s correction (*p* < 0.05) for comparison with positive control and one-way ANOVA test with Tukey’s correction (*p* < 0.05) for multiple comparisons. Instead, data regarding the shelf-life of minced poultry meat and minced rabbit meat (Section 2.4.2, Section 2.4.3, Section 2.4.4, Section 2.4.5 and Section 2.4.6) were examined using a two-way ANOVA test with Sidak’s correction (*p* < 0.05) by comparing each control sample with the corresponding treated sample (P-CT vs. P-TR; YP-CT vs. YP-TR; R-CT vs. R-TR).

## 3. Results

### 3.1. In Vitro Antimicrobial Properties of Pomegranate Extract

PPE shows antimicrobial activity against the tested foodborne pathogenic strains, as evidenced by the diameters of the inhibition zones obtained with the agar-well diffusion method (Figure 1 and Appendix A). In particular, against *E. coli* ATCC 25922, PPE exhibits antimicrobial activity in vitro at a concentration of 10 mg/well, with a mean diameter of the zone of inhibition (MDIZ) of 9.00 ± 1.41 mm, and at 20 mg/well with an MDIZ of 11.00 ± 1.41 mm (Figure 1a). Against *S. aureus* ATCC 25923, the MDIZ is 14.00 ± 1.41 mm at 2.5 mg/well, increasing to 21.00 ± 0.00 mm at a concentration of 20 mg/well (Figure 1b). Against *B. cereus* ATCC 14579, the MDIZ value is 12.50 ± 3.53 mm at a concentration of 2.5 mg/well and 20.50 ± 3.54 mm at a concentration of 20 mg/well (Figure 1c). Against *S. enterica* ATCC 14028, antimicrobial activity at a concentration of 5 mg/well showed an MDIZ of 8.00 ± 0.00 mm, which increased to 11.50 ± 2.12 mm at a concentration of 20 mg/well (Figure 1d).

The in vitro antimicrobial activity of PPE was further confirmed through quantitative assays. MIC and MBC are reported in Table 1. PPE exhibited bacteriostatic and bactericidal effects. The lowest MIC and MBC values were recorded for *B. cereus* ATCC 14579 at concentrations of 5 mg mL^−1^ and 60 mg mL^−1^, respectively. Regarding the other microorganisms, the MIC values were 10 mg mL^−1^ for *S. aureus* ATCC 25923, 20 mg mL^−1^ for *S. enterica* ATCC 14028, and 40 mg mL^−1^ for *E. coli* ATCC 25922, while the minimum bactericidal concentration (MBC) was 80 mg mL^−1^ for all three microorganisms.

### 3.2. Effects of Phytoextract Treatment on Shelf-Life of Minced Meat Samples

#### 3.2.1. Microbiological Analysis of Meat Samples

Shelf-life analysis shows the growth of different microbial groups compared to the limits of satisfaction, acceptability, and dissatisfaction, indicated in Reg. EC 2073/2005 [5], which establishes the methods of analysis for food products placed on the market according to ISO guidelines. In particular, by day 0, the total mesophilic bacteria for all samples fall within the satisfactory range (<10^6^ g^−1^ CFU). On day 3 of storage, a significant reduction is observed in samples of P and R treated with phytoextracts, resulting in 9.4 × 10^7^ CFU g^−1^ and 1.58 × 10^7^ CFU g^−1^, respectively, while YP-treated samples show no significant difference from the control. After day 3, all samples, both treated and untreated, reached dissatisfaction (>10^7^ CFU g^−1^) (Figure 2a). Regarding yeasts, these are already at the acceptability limit on day 0 of analysis (10^4^ CFU g^−1^), showing a considerable increase on days 3 and 6 (Figure 2b). Concerning Enterobacteriaceae, at day 0, all samples are within the satisfactory range (<10^4^ CFU g^−1^), while from day 3 to day 6, an increase beyond the unsatisfactory limits (>10^5^ CFU g^−1^) is recorded for all samples (Figure 2c). *E. coli* β-glucuronidase positive in treated poultry and rabbit meat samples, although showing an increase on day 3, fall within the satisfactory range (<10^4^ CFU g^−1^) until day 6 of analysis; this is in stark contrast to the control samples, where they exceed the unsatisfactory limit (>10^5^ CFU g^−1^) on days 3 and 6 (Figure 2d). Conversely, in all treated samples, there is a decrease in growth. Coagulase-positive Staphylococci are nearly absent, both for control and treated samples, and this condition remains unchanged for poultry- and rabbit-treated samples, even on day 3 and day 6. On the other hand, for control samples, there is an increase within acceptable limits in P and R, while in YP they are not found (Figure 2e). Regarding *Pseudomonas* spp., at day 0, all samples are within the satisfactory range; however, from day 3 to day 6, there is a significant increasing trend. Nevertheless, the treated samples show less statistically significant growth than the controls, achieving an acceptable range until day 6 (Figure 2f). Although not required by the considered regulations, total anaerobic bacteria were analyzed. In particular, these show an increasing trend on days 3 and 6 for all samples. Considering the Clostridium genus in chicken meat samples, they remain within the acceptable range until day 6 (Figure 2g,h).

#### 3.2.2. Biochemical Analysis of Meat Samples

The antioxidant activity in meat samples was evaluated using the DPPH method and the assay was conducted on both raw and cooked meat, as reported in the Materials and Methods (Section 2.4.3). In raw samples (Figure 3a) on day 0 of analysis, compared with control samples, a pronounced antioxidant activity exerted by PPE+OLE can be observed, with values ranging from 57.7% in YP-TR to 70.7% in P-TR and 73.4% in R-TR. The antioxidant activity, observed at days 3 and 6, ranged between 34.6% and 44.5%. After microwave cooking (Figure 3b), P-TR showed an oxidative inhibition percentage of 68.0%, which remained almost stable until day 6 of storage (60.4%). In the cooked YP-TR samples, significant antioxidant activity is recorded on day 0 with a percentage of 59.3%. This activity remains on days 3 and 6 with respective percentages of 45.2% and 31.1%. In the R-TR samples, significant antioxidant activity is observed on day 0 (70%), which decreases to 36.3% on day 3 while remaining statistically significant, and reaches 25.8% on day 6, which is very close to the control sample (15.3%).

The pH evaluation was also conducted on both raw and cooked samples, and the results are reported in Figure 3c and Figure 3d, respectively. At day 0, in both control and treated raw samples, the pH values are between 5.6 and 6.2. During storage, control samples showed an increase in pH on day 3, whereas samples treated with PPE and OLE maintained pH values similar to those of day 0; however, on day 6 of storage, the pH values ranged between 7.4 and 7.7, in all samples. For the cooked samples, no notable variations in pH were observed at day 0, when the pH remained in a range between 5.9 and 6.3. On day 3, pH values increased in all control samples and also in R-TR, although there is a statistically significant difference between R-CT (7.2) and R-TR (6.7). In contrast, on day 3, the pH values of P-TR and YP-TR were almost unchanged compared to day 0. On the final analysis day, an increase in pH value is observed, which becomes slightly alkaline in all control samples. Instead, treated samples show an increase in pH value but they remain within a neutral range (P-TR, 7.2; YP-TR, 6.8; R-TR, 7.1).

The nanomoles of malondialdehyde (MDA) formed from the oxidation of lipids are shown in Figure 3e, where a reduction in MDA in the samples treated with PPE and OLE with respect to the control samples is visible. In particular, P-TR shows a significant decrease in MDA starting from day 0 of storage (0.004 ± 0.002 nmol). This reduction remains significant at day 3 (0.03 ± 0.001 nmol) and day 6 (0.001 ± 0.001 nmol). YP-TR and R-TR show a reduction in nanomoles of malondialdehyde control, although this reduction is not statistically significant.

The color of the meat samples was evaluated using the Computer Vision System technique and the results are shown in Figure 4 and Table 2. It can be noted that P-TR and R-TR showed a more intense yellow coloration than the control, with variations in yellowness (b**^●^**) of about 30 points per analysis (Table 2).

## 4. Discussion

Numerous studies have reported the in vitro antimicrobial activity of pomegranate extracts on human pathogens [10,32,33]. Accordingly, we observed a high antimicrobial activity against all tested strains, although the antimicrobial activity was higher against Gram-positive strains—where the inhibition zone was observed from a concentration of 2.5 mg /well—compared to Gram-negative strains—where antimicrobial activity was observed starting from 5 mg/well and 10 mg/well, respectively, for *S. enterica* ATCC 14028 and *E. coli* ATCC 25922. At a concentration of 10 mg/well, MDIZ values exceeding 8.00 mm were observed against both Gram-positive and Gram-negative bacteria, and MIC values ranged from 5 mg mL^−1^ to 40 mg mL^−1^. These results are in line with those of Wafa et al. on *S. enterica* isolated from chicken meat [34] and also with those of Hanafy et al. [35], who reported that polyphenols extracted from pomegranate peel significantly inhibited food pathogen strains such as *Bacillus cereus*, *Staphylococcus aureus*, *Listeria monocytogenes*, and *Salmonella typhimurium*. In fact, polyphenolic compounds of pomegranate by-products exhibit antibacterial properties, inhibiting bacterial growth by forming complexes with proteins and sulfhydryl groups, rendering these essential components unavailable to bacteria [23]. Simultaneously, they reduce the pH gradient around the cell membrane, leading to the depletion of ATP and eventual microbial cell death [36]. Furthermore, punicalagin, one of the main active compounds in pomegranate peel, induces morphological damage to the cell membrane and exerts a notable inhibitory effect on the formation of microbial biofilms [37]. Protein expression has also recently been demonstrated as an assailable bacterial target of punicalagin [38].

Meat and meat by-products are prone to degradation during storage due to microbial activity and chemical reactions, such as protein and lipid oxidation, which limits their shelf-life. Their high nutrient content also makes them particularly susceptible to microbial contamination. Our results on poultry and rabbit meat products confirm the antimicrobial activity of PPE against nearly all the microorganisms investigated up to 6 days of refrigerated storage. In particular, we highlighted a decrease in Enterobacteriaceae in the samples treated, specifically regarding *E. coli* β-glucuronidase-positive strains where a significant reduction was observed, reaching acceptable limits by days 3 and 6. Similarly, a decrease in Pseudomonaceae was observed in all treated samples, with P-TR and R-TR even falling below the satisfactory limit on days 3 and 6 in line with Kanatt et al. [39]. It is also interesting to note that coagulase-positive Staphylococci were not detected in the treated samples. Rasuli et al. [17] demonstrated the ability of pomegranate peel extract to maintain the quality of buffalo meat for storage periods of up to 8 days. In fact, antimicrobial compounds present in pomegranates, such as punicalagin and organic acids, likely contribute to inhibiting both Gram-positive and Gram-negative bacteria in the meat, thereby delaying bacterial growth. Furthermore, fresh meat-based products are easily susceptible to protein and lipid oxidation during the storage period at 4 °C. Oxidative changes and microbiological activity can adversely affect the sensory, stability, nutritional, and acceptability properties of meat products [40]. For these reasons, antioxidants and antimicrobials are increasingly important additives in the meat industry to extend shelf-life and enhance acceptability. Mediterranean plant extracts and essential oils have been investigated as potential natural antimicrobials and antioxidants added to meat and meat products, offering medicinal and functional properties [41]. In recent years, plant derivatives have gained prominence in minimally processed products due to their high phenolic content, which helps enhance color stability and reduce off-flavors [41]. The use of natural compounds is prompted by consumer concerns about the presence of synthetic additives commonly used in meat product processing, such as nitrites, sulphites, butylated hydroxytoluene (BHT), or butylated hydroxyanisole (BHA), and their willingness to purchase healthier and safer meat alternatives [24]. Different parts of the *Olea europaea* contain antioxidant compounds. Numerous researchers have applied DPPH and ABTS radical assays to evaluate the radical scavenging capacity and characterize olive processing by-products’ antioxidant activity [19,42,43]. Recent studies have investigated the impact of *Olea europaea* polyphenols on fresh and minced meat, as well as on meat products including patties, frankfurters, deep-fried cuts, and dry-fermented sausages [24]. Aouidi et al. [43] utilized olive extract at concentrations of 100–150 µg g^−1^ of minced meat, noting a significant decrease in lipid oxidation during refrigerated storage. Considering the obtained in vitro antimicrobial results on foodborne pathogens, the literature on the antimicrobial activity of pomegranate peel extract used for food preservation, and the antioxidant activity of olive leaf extract in meat [17,25,39,44], the concentrations of 10 mg of PPE and 0.25 mg of OLE for gram of minced meat in combination were selected for the treatment of chicken and rabbit meat to improve their shelf-life, in order to evaluate the reduction in microbial growth over time and to assess the biochemical and nutritional changes in meat products.

As mentioned before, polyphenols can remove free radicals through scavenging activity. In fact, the percentage of oxidative inhibition evaluated through the DPPH assay indicates a strong antioxidant activity for all treated samples, both cooked and raw; however, this activity decreases during storage, particularly on day 3, and then remains relatively constant on day 6. Several studies report that polyphenols exhibit potent antioxidant effects, beneficial for neutralizing reactive oxygen species, thereby protecting cells and tissues [45,46,47,48]. The biochemical analysis of thiobarbituric acid reactive species (TBARS) reveals that, among the examined samples, the treated ones demonstrate lower lipid peroxidation levels, indicating more effective preservation of unsaturated fatty acids. Other studies observed a significant reduction in the levels of substances reactive to thiobarbituric acid—as well as inhibition of lipid oxidation and the production of secondary oxidation products—when utilizing pomegranate by-products in sausages and olive leaf extract in raw rabbit meat, chicken Frankfurters, and poultry meat [25,47,48,49,50].

On a molecular level, the major phenolic components of OLE exhibit antioxidant properties due to their catecholic structure. They achieve this in two main ways: first, by scavenging peroxyl radicals and interrupting peroxidative chain reactions, thereby forming very stable resonance structures [51]; second, by acting as metal chelators, which prevent the oxidation of low-density lipoproteins induced by copper sulphate [52]. The metal-chelating activity of OLE can be attributed to the electron-donating ability of the hydroxyl groups and the resulting formation of intramolecular hydrogen bonds with free radicals [53].

Similarly, pH monitoring is one of the important physico-chemical parameters used to determine freshness, influencing the quality and shelf-life of meat products. In raw samples, slight acidification is observed in all treated samples compared to the controls; however, this difference diminished on day 6, where the pH of all samples ranged between 7.3 and 7.5. In cooked samples, this statistically significant decrease in pH was evident on day 3 and persisted for all treated samples on day 6. Devatkal et al. [54] also observed a pH reduction in pomegranate-treated meatballs, likely attributed to the acidic nature of pomegranate itself. At the same time, Morsy et al. [44] observed that treating meatballs with pomegranate peel nanoparticles prevented the increase in pH during storage, likely caused by the decomposition of nitrogenous compounds by endogenous or microbial enzymes. Similarly, Saleh et al. observed a smaller pH variation in poultry meat samples treated with olive leaf extract [35]. As reported in the literature, the treatment with polyphenols reduces slightly the pH values in meat samples, helping to prevent microbial growth [17,55]. Although treatment with polyphenols leads to a reduction in microbial growth and lipid oxidation without affecting the pH, which remains in the neutral range, it causes a colorimetric variation in the product. This variation may be due to the color of the pomegranate extract itself and not to the rancidity of the meat. In fact, in yellow poultry (YP) samples, adding the extract has no significant impact on color.

Various factors influence the interactions between polyphenols and other food compounds, including the chemical characteristics of the food species involved, pH levels, and temperature [56]. The literature on PPE and OLE limitations as food additives is quite scarce. Few studies have explored the complex interactions between polyphenols and other food compounds, with conflicting results. The interaction of polyphenolic compounds, used as food additives, with the macromolecules in a food matrix can reduce the biological properties of the polyphenols (such as antioxidant and antimicrobial effects) and affect the availability, digestibility, and absorption of the macromolecules [57]. For instance, pomegranate is noted for its astringent effect when used in food systems [58]. Additionally, the solubility of polyphenols in fatty food matrices has been addressed by only a very limited number of studies [59]. Encapsulation in suitable materials could enhance their solubility and mask any unpleasant flavors [60], thereby broadening the application of phytoextracts from pomegranate and olives as food additives. Consequently, further studies on sensory analysis and consumer acceptance of food products containing PPE and OLE are warranted.

## 5. Conclusions

Polyphenols derived from agri-food industry waste are widely used as natural preservatives and dietary supplements, also leading to the reuse of production waste and aligning with a green production policy. The combination of PPE and OLE has been shown to improve the chemical and microbiological quality of poultry and rabbit minced meat. Both PPE and OLE are a potent source of polyphenols with antioxidant and antimicrobial properties that can inhibit microbial growth and counteract lipid oxidation and deterioration of meat products. These natural substances, in addition to being approved by the EFSA, also benefit the consumer. In fact, it has been demonstrated that olive leaf extract (OLE) can modulate gut microbiota, promoting probiotic bacteria, thereby enhancing health benefits for humans [61], while pomegranate extract has antioxidant activity in the cells of the gastrointestinal tract [62]. Therefore, their use as food preservatives can not only improve the shelf-life of easily perishable, fresh products such as chicken and rabbit meat but also add nutritional value to the product with beneficial effects on the consumer’s health. Moreover, it is crucial to develop cost-effective techniques for extracting the bioactive components of pomegranate and olive for large-scale application. Implementing these methods will not only enhance our understanding of how to use the phytocompounds derived from agro-industrial waste but also advance food science research, benefiting both consumer health and the environment.

## Figures and Tables

**Figure 1 microorganisms-12-01303-f001:**
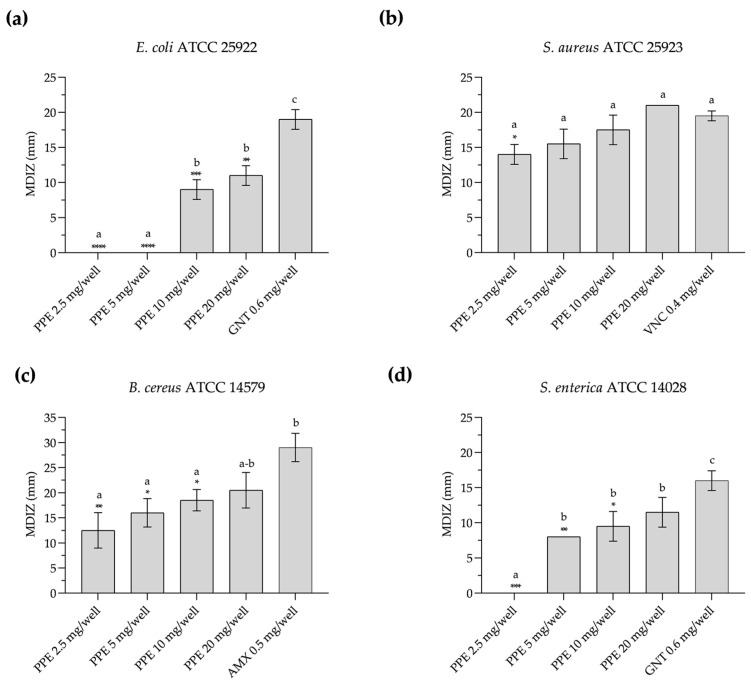
In vitro antibacterial activity of pomegranate peel extract evaluated by the agar-well diffusion method against *E. coli* ATCC 25922 (**a**), *S. aureus* ATCC 25923 (**b**), *B. cereus* ATCC 14579 (**c**), and *S. enterica* ATCC 14028 (**d**). The mean diameter of the inhibition zone (in mm) is reported as the mean of values obtained from assays in triplicate ± standard deviation. Statistical significance was examined by a one-way ANOVA test with Dunnett’s correction (*p* < 0.05) for comparison with positive control and a one-way ANOVA test with Tukey’s correction (*p* < 0.05) for multiple comparisons. Asterisks indicate statistical significance with respect to positive control (**** *p* < 0.0001; *** *p* < 0.001; ** *p* < 0.01; * *p* < 0.05); absence of asterisks indicates absence of significance. Letters (a–c) indicate statistical differences between different values; results with no significant differences receive the same letter. MDIZ, mean diameter of inhibition zone; PPE, pomegranate peel extract; GNT, gentamicin; VNC, vancomycin; AMX, amoxicillin.

**Figure 2 microorganisms-12-01303-f002:**
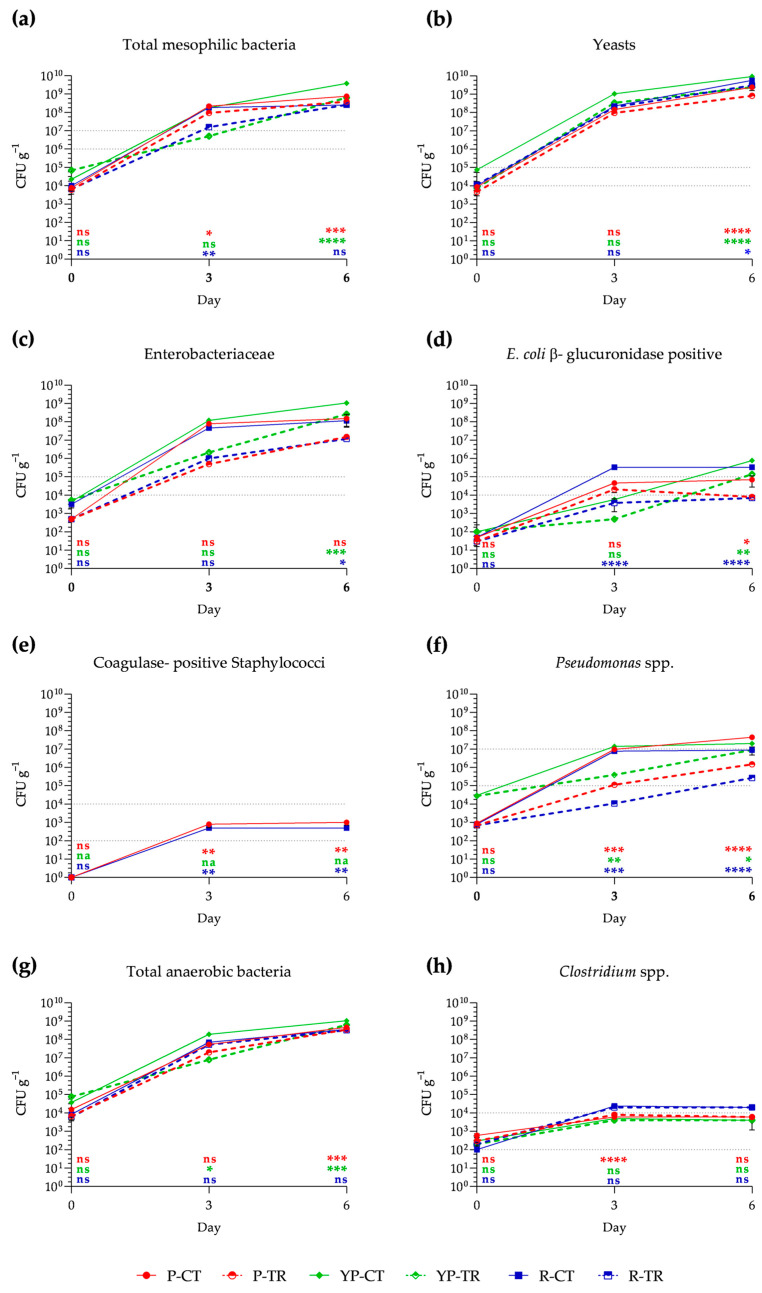
Growth of total aerobic mesophilic bacteria (**a**), yeasts (**b**), Enterobacteriaceae (**c**), *E. coli* β-glucuronidase positive (**d**), coagulase-positive Staphylococci (**e**), *Pseudomonas* spp. (**f**), total anaerobic bacteria (**g**), and *Clostridium* spp. (**h**) in poultry (P), yellow poultry (YP), and rabbit (R) minced meat samples at different storage times (0, 3, 6 days) at 4 °C, in control (CT) samples and treated (TR) samples with PPE and OLE. The mean values ± SD, obtained from triplicate analyses, are expressed as Colony-Forming Units (CFU) per gram of meat analyzed. Gray dashed lines represent the acceptability limits established by CE Regulation 2073/2005 for refrigerated minced meat: the bottom line represents the limit for satisfactory; the acceptability range falls within the two lines; while the top line represents the limit of unacceptability. Statistical significance was examined using a two-way ANOVA test with Sidak’s correction (*p* < 0.05) by comparing each control sample with the corresponding treated sample (P-CT vs. P-TR; YP-CT vs. YP-TR; R-CT vs. R-TR). Statistical analysis is reported for each group with the same color (P: red; YP: green; R: blue). Asterisks indicate statistical significance with respect to the control (**** *p* < 0.0001; *** *p* < 0.001; ** *p* < 0.01; * *p* < 0.05). The abbreviations are as follows: P-CT (poultry control or untreated); YP-CT (yellow poultry control or untreated); R-CT (rabbit control or untreated); P-TR (poultry treated with polyphenol mixture); YP-TR (yellow poultry treated with polyphenol mixture); R-TR (rabbit treated with polyphenol mixture); ns, not significant; na, not available.

**Figure 3 microorganisms-12-01303-f003:**
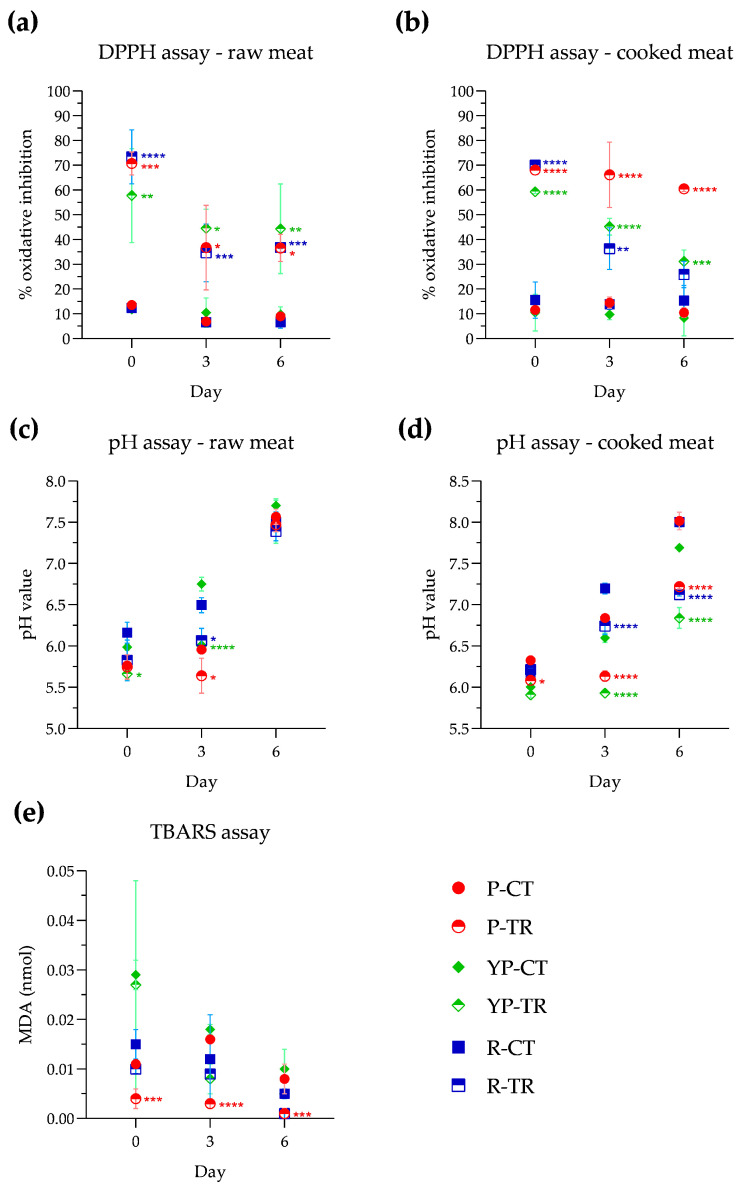
Evaluation of antioxidant activity in raw meat (**a**) and cooked meat (**b**); pH value of samples raw (**c**) and cooked meat (**d**); and lipid peroxidation expressed as nmol of malonyldialdeide (MDA) (**e**) in poultry, yellow poultry, and rabbit minced meat at different storage times (0, 3, 6 days) at a temperature of 4 °C, in the absence and presence of treatment with PPE and OLE extracts. Statistical significance was examined by two-way ANOVA test with Sidak’s correction (*p* < 0.05) by comparing each control with the corresponding treated samples (P-CT vs. P-TR; YP-CT vs. YP-TR; R-CT vs. R-TR). Asterisks indicate statistical significance between the samples (**** *p* < 0.0001; *** *p* < 0.001; ** *p* < 0.01; * *p* < 0.05). Statistical analysis is reported in each graph with the same identifying colors of the samples (P: red; YP: green; R: blue). P-CT (poultry control or untreated); YP-CT (yellow poultry control or untreated); R-CT (rabbit control or untreated); P-TR (poultry treated with phytoextracts); YP-TR (yellow poultry treated with phytoextracts); R-TR (rabbit treated with phytoextracts).

**Figure 4 microorganisms-12-01303-f004:**
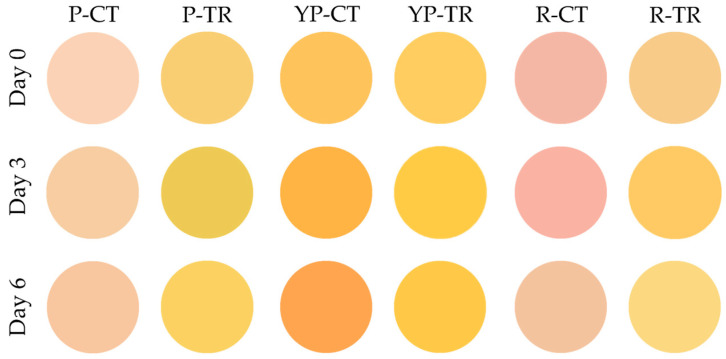
Computer Vision System evaluation of meat color. P-CT (poultry control or untreated); YP-CT (yellow poultry control or untreated); R-CT (rabbit control or untreated); P-TR (poultry treated with phytoextracts); YP-TR (yellow poultry treated with phytoextracts); R-TR (rabbit treated with phytoextracts).

**Table 1 microorganisms-12-01303-t001:** Quantitative evaluation of in vitro antibacterial activity of PPE.

Microorganisms	PPE [mg mL^−1^]		GNT [µg mL^−1^]		VNC [µg mL^−1^]		AMX [µg mL^−1^]	
	MIC	MBC	MIC	MBC	MIC	MBC	MIC	MBC
*E. coli* ATCC 25922	40	80	4	10	nt	nt	nt	nt
*S. aureus* ATCC 25923	10	80	nt	Nt	1.5	2.5	nt	nt
*B. cereus* ATCC 14579	5	60	nt	Nt	nt	nt	50	200
*S. enterica* ATCC 14028	20	80	25	100	nt	nt	nt	nt

PPE, pomegranate peel extract; GNT, gentamicin; VNC, vancomycin; AMX, amoxicillin; MIC, minimum inhibitory concentration; MBC, minimum bactericidal concentration; nt, not tested.

**Table 2 microorganisms-12-01303-t002:** Instrumental color values of meat products—expressed as mean ± SD of lightness, redness, and yellowness—using a Computer Vision System.

		P-CT	P-TR	YP-CT	YP-TR	R-CT	R-CT
Day 0	L**^●^**	86 ± 2	84 ± 8	82 ± 6	84 ± 6	78 ± 8	83 ± 4
	a**^●^**	8 ± 3	4 ± 9	12 ± 3	6 ± 8	16 ± 7	6 ± 4
	b**^●^**	18 ± 10	46 ± 14	55 ± 7	55 ± 3	16 ± 11	36 ± 14
Day 3	L**^●^**	84 ± 2	82 ± 6	78 ± 2	84 ± 3	77 ± 4	83 ± 3
	a**^●^**	7 ± 3	0 ± 9	20 ± 6	8 ± 5	19 ± 8	7 ± 5
	b**^●^**	25 ± 13	56 ± 13	60 ± 3	64 ± 3	16 ± 16	52 ± 16
Day 6	L**^●^**	82 ± 3	85 ± 5	73 ± 5	83 ± 4	81 ± 6	87 ± 8
	a**^●^**	10 ± 5	2 ± 6	21 ± 7	8 ± 5	10 ± 4	2 ± 10
	b**^●^**	24 ± 10	55 ± 12	50 ± 9	62 ± 7	23 ± 12	44 ± 10

P-CT (poultry control or untreated); YP-CT (yellow poultry control or untreated); R-CT (rabbit control or untreated); P-TR (poultry treated with phytoextracts); YP-TR (yellow poultry treated with phytoextracts); R-TR (rabbit treated with phytoextracts); L**^●^**, lightness; a**^●^**, redness; b**^●^**, yellowness.

## Data Availability

Data are contained within the article.

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
