# Peer review of "Pomegranate Peel and Olive Leaf Extracts to Optimize the Preservation of Fresh Meat: Natural Food Additives to Extend Shelf-Life"

_microorganisms, 2024, doi:10.3390/microorganisms12071303_

Round 1
Reviewer 1 Report
Comments and Suggestions for Authors
The originality of the study and the novelty it brings to the field is of actuality. The purpose of the article and its significance are stated clearly. The paper is well-structured, the abstract is concise, and the topic; in the introduction is supported by well-selected bibliographic data.
The manuscript could be improved by studying other papers in the field, I kindly recommend the next paper to be consulted https://doi.org/10.3390/plants12010131
The beneficial effects of PPE and OLE are mentioned. Still, the manuscript does not explain the underlying mechanisms in detail, such as how these polyphenols specifically inhibit microbial growth and lipid oxidation.
The text does not mention any potential limitations or downsides of using PPE and OLE as preservatives, such as possible interactions with other food components, sensory changes in the meat, variability in effectiveness depending on storage conditions, or practical challenges associated.
At the same time, it is important to state clearly the implications for research, practice, and society.
Reviewer 2 Report
Comments and Suggestions for Authors
The manuscript titled with " Pomegranate peel and olive leaf extracts to optimize the preservation of fresh meat: natural food additives to extend shelf-life ". The manuscript discusses a good point. Overall, the presented study indicates for the antimicrobial activity of pomegranate peel extract and the antioxidant activity of olive leaf extract, added at respectively, to minced poultry and rabbit meat as food additives to preserve the quality and extend the shelf-life of meat products. The manuscript written well. But it needs a minor revision (Minor editing of English language required)
Comments on the Quality of English LanguageMinor editing of English language required
